# Survival of immature pre-adult *Gnathostoma spinigerum* in humans after treatment with albendazole

**Tapanee Kanjanapruthipong[1], Sumate Ampawong[1], Urusa Thaenkham[2], Khwanchanok Tuentam[1], Dorn Watthanakulpanich**[2]*

1 Faculty of Tropical Medicine, Department of Tropical Pathology, Mahidol University, Bangkok, Thailand,
2 Faculty of Tropical Medicine, Department of Helminthology, Mahidol University, Bangkok, Thailand

* dorn.wat@mahidol.edu

**Data Availability Statement:** All relevant data are within the paper and its Supporting Information files.

## Abstract

Human gnathostomiasis is a food-borne zoonotic helminthic infection widely reported in Latin America, Asia and Southeast Asia, particularly in Thailand. There are increasing reports of the parasite in countries where it is not endemic. A study of the survival drug-treated immature stage (STIM) of *Gnathostoma spinigerum* recovered from infected patients focused on their integument surface using scanning electron microscopy (SEM). STIM displayed a specific, characteristic head bulb, with a pair of large thick equal-sized tri-lobulated lips in the centre. Cephalic spines had eight transverse rows on the head bulb with single-ended tips curved posteriorly. Body cuticular spines on the anterior half of the STIM were not sharp-pointed but distributed more densely, with multi-dentated-cuticular spines, irregularly arranged in a lining pattern of velvety cuticular folds. The length of cuticular spines increased caudally. The size of spines became gradually smaller, and numbers decreased towards the posterior end. Spines were still widely dispersed posteriorly as their density dropped. The morphology of STIM of *G. spinigerum* are described in detail for the first time. These specimens showed structural adaptation based on changes on integument surfaces, probably to protect against damage induced by the toxic effects of albendazole.

## Introduction

Human gnathostomiasis is endemic in Thailand and is caused by the random migration of only *G. spinigerum* larvae. According to the summary of the Immunodiagnostic unit under the Department of Helminthology, Faculty of Tropical Medicine, Mahidol University, the infection rates of gnathostomiasis from positive results of western blot analysis were 44.5%, 35.0% and 33.4% evaluated during 1995–2005, 2006–2009, 2019–2021, respectively (Unpublished data). These immature helminths cannot develop into mature adult worms, and humans are accidental and dead-end hosts [1–5]. The most common clinical manifestation is associated with advanced third-stage larvae (AL3). Helminths in this stage are localised in intermittent cutaneous migratory swellings. These swellings are associated with localised piecing pain,

**Funding:** The author(s) received no specific funding for this work.

**Competing interests:** The authors have declared that no competing interests exist.

**Abbreviations:** SEM, Scanning electron microscopy; SPB, Sucrose phosphate buffer.

pruritus and erythema that persist for years. These symptoms are related to mechanical disruption caused by larval migration [3,6]. Erratic migration of larvae to various organs results in larval migration syndrome in the skin, subcutaneous tissues, liver, lungs, kidneys, brain and other tissues [7–10].

No completely effective anthelminthic treatment for this helminthic infection is currently available. However, benzimidazoles, including albendazole, broad-spectrum anthelminthic derivatives, are widely used. This drug is effective for intestinal helminthiasis. Its mechanism of action is an attack on the structural cytoskeletal protein and tubulin. The drug binds to helminth microtubules with subsequent depolymerisation. This action results in a blockage of glucose uptake in many parasitic nematodes [11]. Albendazole is rapidly and extensively metabolised into therapeutically active albendazole-sulphoxide (AlbSO) in the liver [12]. In general, albendazole is prescribed at 400 mg twice per day or 800 mg once a day in either two-week or three-week courses. Cure rates are as high as 78.5%–94.0%, as is a recurrence of migratory swelling symptoms [13–15]. Relapses are attributed to treatment failure rather than reinfection [16].

Little information on STIM after treatment is available due to difficulties in collecting surviving worms from patients. These helminths must be drug-resistant and survive inside the body. Further, they must migrate to organs, particularly subcutaneous tissue or skin, where they are visible. Finally, patients need to recognise outward migration after treatment before they visit the clinic. There are no earlier studies on morphological changes in surviving *G. spinigerum* after treating human patients with albendazole. Hence, this study's results will increase understanding of integument changes characteristic of STIM.

## Material and methods

### Ethics approval

The Ethics Committees of the Faculty of Tropical Medicine, Mahidol University for ethical clearance processing approved this study (No. TMEC 21–033). The Faculty of Tropical Medicine Institutional Animal Care and Use Committee approved the animal studies (approval number U1-02745-2559).

### Patient profiles

This study included all patients who visited clinicians at the Parasite Clinic of the Hospital for Tropical Diseases, Faculty of Tropical Medicine, Mahidol University between 2013 and 2019 with a chief complaint of intermittent migratory circumscribed swelling. A total of 476 patients—180 males and 296 females participated. Patients were informed to carefully observe any abnormal indurated, erythematous plaques [17] or an associated migratory erythematous rash in subcutaneous tissue or superficial skin on any part of the body after treatment with albendazole. Some gnathostome larvae were stimulated to migrate outwardly toward the skin surface and were visible. Worms were removed by picking with a sterile needle [14,18]. Only three patients showed the extremely rare resistant *G. spinigerum* worms. **Table 1** provides detailed patient information.

### Pathogens

The general and structural morphology of three *G. spinigerum* STIM specimens were studied in detail using scanning electron microscopy (SEM). Cephalic and cuticular spines and caudal sections of the body were assessed and compared with control (free from albendazole) specimens of L3, AL3 and fully grown adult worms.

**Table 1. Clinical data for three female gnathostomiasis patients who were the source of three recovered STIM.**

| Genders Year | Ages | Habits | Underlying diseases | Habitat location | Vital signs | Swelling organs | Laboratory investigations | Treatments |
|---|---|---|---|---|---|---|---|---|
| Female 2015 | 28 | Favoured various improperly cooked meals | Thyrotoxicosis | Ubon Ratchathani Province Thailand | BW 62.5 BP 121/71 PR 82 RR 20 | Gastrocnemius with itching sensation for 3 days | Stool exams: Parasite not found. Western blotting positive reacting band at 24 kDa | Albendazole (200 mg) 2x2 |
| Female 2018 | 19 | Refused raw cooked meals but favoured having fried catfish, snake-headed fish | None | Myanmar | BW 47.2 Ht 138.5 BP 1o4/89 PR 80 RR 20 | Left wrist with itching sensation twice a year for about 2 years | Stool exams: Parasite not found. Western blotting positive reacting band at 24 kDa | Albendazole (200 mg) 2x2 |
| Female 2020 | 30 | Favoured fermented fish (anchovy), frogs, crabs | Hypertension | Kalasin Province Thailand | BW 41.5 Ht 156.0 BP 93/68 PR 84 RR 20 | Right thigh radiated to right inguinal area; off and on for 1 year | Stool exams: Parasite not found. Western blotting positive reacting band at 24 kDa | Albendazole (200 mg) 2x2 |

**G. spinigerum third-stage larvae (L3) from Asian swamp eels.** L3 were collected from cysts in the livers of dead Asian swamp eels (*Monopterus alba*) bought at a market in Bangkok. Internal viscera normally removed and discarded were collected and transferred to the laboratory at the Department of Helminthology, Faculty of Tropical Medicine, Mahidol University. Livers were isolated from gastrointestinal tracts, pooled and digested at 37°C for 3 hours in artificial gastric fluid containing 0.7% pepsin (1:10,000, Sigma-Aldrich Co., St. Louis, Missouri) and 1.0% HCl. The digested mixture was washed with 0.85% NSS several times until the mixture became clear. Gnathostome larvae were collected by simple sedimentation, and active larvae were identified under a dissecting microscope. These larvae were used for comparisons with survival worms.

**G. spinigerum advanced third-stage larvae (AL3) from a Wistar rat.** Twenty AL3 collected from Asian swamp eels were fed to a Wistar rat using an 18-gauge oral gavage needle. Seventeen days post-infection, the animal was euthanised in a $CO_2$ chamber. The whole rat body was examined by tissue compression to recover gnathostome larvae from cysts in muscle or liver under a stereomicroscope. Recovered larvae were then washed and deformed in 0.85% normal saline.

**G. spinigerum fully grown adults from stray dogs.** Two pairs both males and females of *G. spinigerum* adults were a gift from Prof. Prayong Radomyos and used as controls. The worm adults were taken from stray dogs (mixed breed) without proven known age information in Bangkok.

**G. spinigerum immature young adults from patients.** Three STIM were recovered from three patients who tested positive for human gnathostomiasis in confirmatory Western blotting [3,7,19]. General morphology identified species recovered from infected patients by number, form and distribution of cephalic spines, including numbers in each row, shape and distribution of body cuticular spines, and body sizes [20–22]. The worms were consistent with an immature young adult stage of *G. spinigerum* and were kept as referenced materials in the Department of Helminthology, Faculty of Tropical Medicine, Mahidol University.

## Laboratory investigations

**Permanent mount.** Detailed morphology of two fully grown *G. spinigerum* adults both male and female each was assessed after fixing in 10% neutral formalin under coverslip pressure, clearing in alcohol-glycerin solution and mounting in glycerin jelly. Morphology identified important parasite characteristics.

**Permanent stain.** Five L3 and one STIM were fixed in AFA (95% Ethyl alcohol, Formaldehyde, Glacial acetic acid) solution for 1 hour and immersed in 70% ethanol for 5 minutes. The worms were then put in Carmine stain for 24 hours until they turned pink. The worms were then washed with 1% HCO alcohol and dehydrated by soaking in 70% and 95% ethanol for 15 minutes to remove any remaining colour. Finally, the L3 were counter-stained with Fast green for 2–3 seconds. Clearing used immersion in absolute ethanol and xylene infused with absolute ethanol for 15 minutes each, then xylene again for 10 minutes. The specimens were then permanently mounted.

**Scanning electron microscopy.** Ten L3, 10 AL3, 2 fully-grown adults and 2 STIM were fixed in 2.5% glutaraldehyde in 0.1 M sucrose phosphate buffer (SPB) for 1 h. Fixed worms were washed in 0.1 M SPB and fixed with 1% osmium tetroxide in 0.1 M SPB for 1 h. All gnathostome worms were then dehydrated in graded ethanol solutions and dried in liquid $CO_2$ using a critical point dryer machine (HITACHI HCP-2, Japan). After that, each worm was mounted on an aluminium stub using double-sided carbon tape and coated with a 20 nm gold film using a sputter coater (EMITECH K550, UK). The STIM specimens were examined under SEM (JEOL JSM-6610LV, Japan) with 15 kV acceleration voltages for comparison with L3, AL3 and fully-grown adults as controls.

## Statistical analysis

Lengths and numbers of spines from each gnathostome stage were compared by one-way analysis of variance using SPSS programme, version 21, and expressed as means ± SD. The 95% confidence interval, $p < 0.05$, was considered the threshold for statistical significance.

## Results

### General morphology of the STIM

Viable STIM continually moved in saline for several hours after removal from patients. The specimen was short, stout and transparent with a pinkish colour. Size was $5,288.00 \times 658.40$ μm (**Fig 1**). Two equally thick trilobulated lips appeared at the anterior end. There were eight transverse rows of cephalic spines (thick arrow) and a semispherical head bulb $245.70 \times 435.60$ μm in length and width. The bulb was easily distinguishable from the body. Internal organs, such as cervical sacs, oesophagus and intestine were seen (thick arrow). In addition, the intestine is connected to the oesophagus, straight and hollow down to the anal opening near the posterior end. Numerous crooked and wicked cuticular spines, with velvety cuticular folds were distributed on the anterior one-third of the body and these structures were not found in L3, AL3 and fully-grown adult control specimens (**Figs 1 and 2**). The intestine of the STIM occupied most of the body cavity and displayed an appearance like fully grown adults in both male and female permanent mounts (thick arrow) (**Fig 2D and 2E**). The STIM was identified as a stage of development beyond AL3 with well-developed cuticular spines on the entire body and size consistent with an immature male adult. Cephalic spines were counted and measured under SEM for comparison with controls (**Table 2**). However, there was also areas without cuticular spines observed connected to the single-pointed spines for the STIM. On the ventral side of the tail, the STIM presented minute unidentate spines and

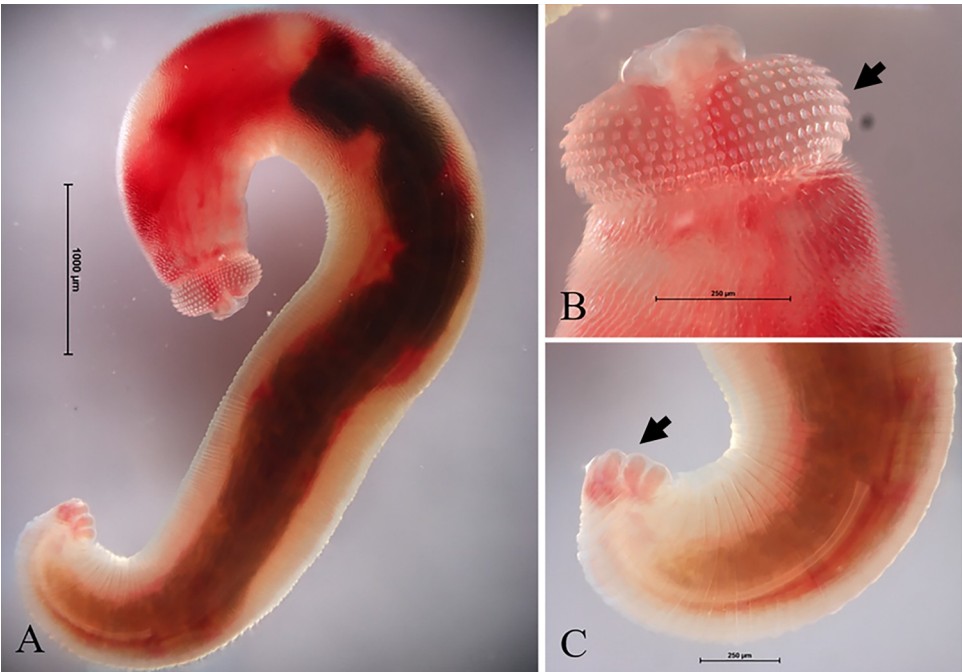

**Fig 1. Light micrograph of permanent stain of STIM.** (A); Head bulb with eight rows of cephalic hooklets (B); Caudal area (C).

four pairs of large tongue-shaped caudal papillae (thick arrow) but without the single pair of unequal reproductive spiculae generally found in adult males (**Figs** 1 and 2H).

## SEM study of the STIM

Under SEM, the STIM worm each showed a head bulb easily distinguished from the body. Head bulb and body size increased from L3 to AL3 to STIM to fully-grown adult. The STIM mouth showed a pair of thick, equal, half-moon-shaped trilobulated lips. The head bulb displayed several well-developed single-ended triangular tips with irregular polygonal or oblong bases of almost equal size and armed with eight transverse rows and the average number of similar cephalic spines were 38, 46, 50, 60, 62, 62, 60 and 32, respectively (**Fig 3**, **Table 2**). The curved line on the head bulb of the L3 specimen showed a smoother, obtuse angle (arrow head) accompanied by continuous thick, well-developed oblong-shaped single-pointed spines in rows of more than 40 cephalic spines (**Fig 3A**). The length of each cephalic spine was greater than in AL3, STIM and adult stages. The latter group had more cephalic spines that developed later and appeared sharp-pointed and backwardly directed in rows with a similar claw-like shape (arrow head) (**Fig 3B–3D**).

The STIM's entire body surface was covered with body cuticular spines in rows from the cervical constriction to the posterior end and arranged in about 885 rows with transverse striations up to the posterior extremity. The average interval between transverse striations was 12.45, 7.96 and 3.39 μm for the anterior, middle and posterior parts of the body, respectively. These measurements are less than recorded for fully grown adults; 15.56, 8.65 and 5.14 μm, with the larger body size. Abundant crooked and wicked cuticular spines separated by narrow spaces were distributed over 35% of the body. The anterior half of the STIM specimen was densely covered with multi-dentated cuticular spines similar to spines of the fully grown mature adult. The spines were irregularly in a lining pattern of velvety cuticular folds (arrow

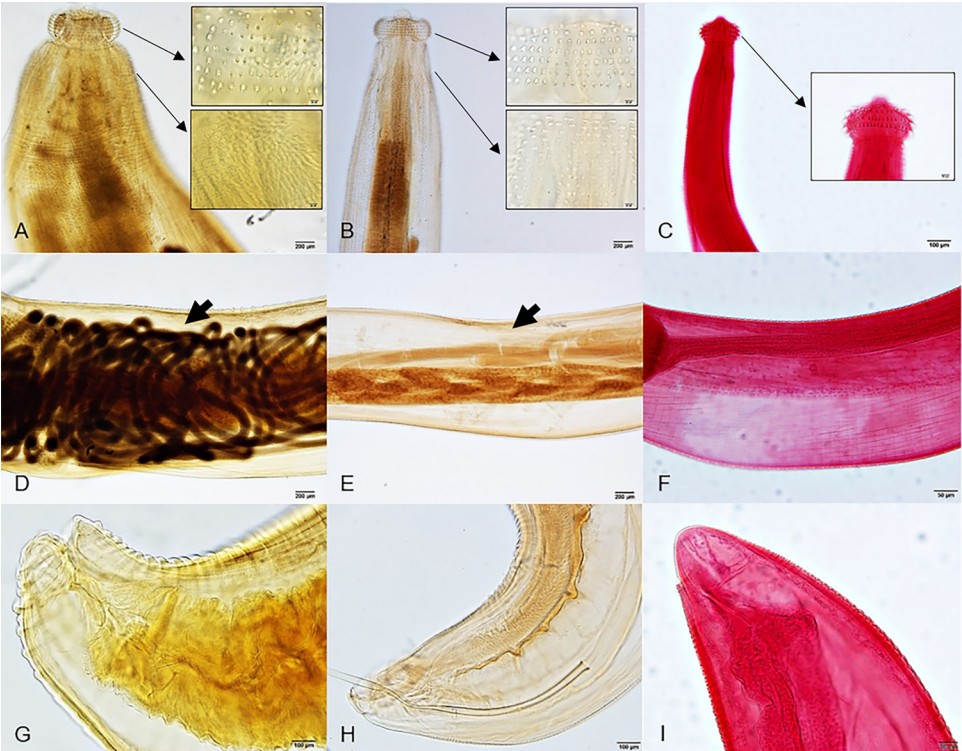

**Fig 2. Light micrographs of permanent mount of Adults and permanent stain of AL3.** Adult female (A, D, G) and male (B, E, H) with anterior (A), middle (D) and caudal (G) areas, respectively. Anterior cephalic hooklets and cuticular spines are shown in inset images (A-C).

head) (**Fig 3H**). In contrast, the L3 (**Fig 3E**) and AL3 forms (**Fig 3F**) showed clear striation with transverse rows of body cuticular spines. Ellipsoidal cervical papillae without spines varied between rows 9 and 10 in the larval stage of L3 and AL3 to between rows 20 and 22 of cuticular spines in more developed pre-adult and adult stages (**Fig 3**). The body cuticular spines

**Table 2. Comparative morphological characteristics of the STIM specimens with other stages of *G. spinigerum* under scanning electron microscopy.**

| Area | Rows | Magnitude of spine (μm; mean ± SD) (Number of spines) | | | |
|---|---|---|---|---|---|
| | | L3 | AL3 | Fully grown adult | STIM |
| Cephalic spines | 1st | 2.25±0.51 (43) | 1.64±0.29 (44) | 1.25±0.18 (40) | 0.61±.012 (38) |
| | 2nd | 3.25±0.60 (47) | 2.59±0.30 (50) | 1.89±.024 (52) | 0.94±2.06 (46) |
| | 3rd | 3.46±0.46 (49) | 3.01±0.40 (54) | 1.87±0.17 (60) | 1.32±0.16 (50) |
| | 4th | 3.56±0.30 (53) | 2.61±0.43 (58) | 2.21±0.39 (66) | 1.59±0.24 (60) |
| | 5th | | | 2.53±0.22 (64) | 1.58±0.15 (62) |
| | 6th | | | 2.64±0.40 (70) | 1.33±0.17 (62) |
| | 7th | | | 2.37±0.51 (62) | 1.09±0.25 (60) |
| | 8th | | | 1.25±.021 (40) | 0.74±.010 (32) |
| Cuticular spines | anterior | 7.40±0.90 | 3.72±0.46 | 7.58±0.77 | 2.23±.036 |
| | posterior | 1.30±0.31* | 1.23±0.22* | 0.90±0.15* | 3.92±0.5* |
| Body size (Length x Width) | | 2,688 x 338.26 μm | 2,250 x 195.32 μm | 1.5 cm x 1203.76 μm | 5,288 x 658.37 μm |
| Transverse rows of body cuticular spines | | 193 | 312 | 885 | 1,629 |

* $p < 0.05$.

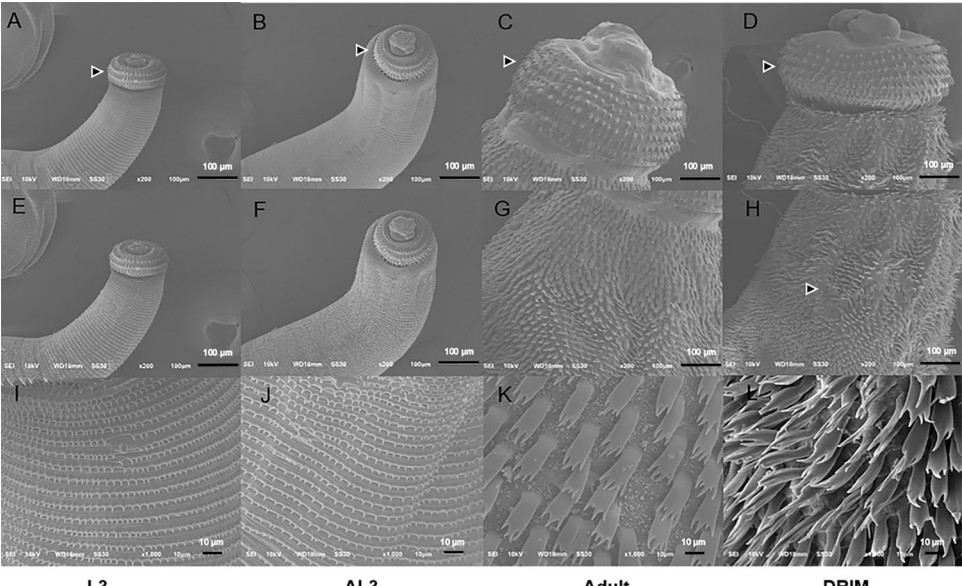

**Fig 3. Electron micrographs of the anterior of *G. Spinigerum*.** Head (A-D); Neck (E-H); Cuticular spines with higher magnification (I-L); L3 (A, E, I); AL3 (B, F, J); adult (C, G, K); STIM (D, H, L).

were larger and more densely distributed anteriorly and gradually smaller in numbers and sizes with wide dispersion posteriorly in all stages.

The pattern of body cuticular spines close to the head bulb of the STIM form was comb-like with sharp three-pointed ends (trifurcated spines). The longest middle-point spines gradually became two-pointed (bifurcated spines) at a level near the middle of the worm body. This morphology continued as far down as 50% of the body length where they again became longer with tiny single-pointed ends. Statistically significant differences between STIM and controls

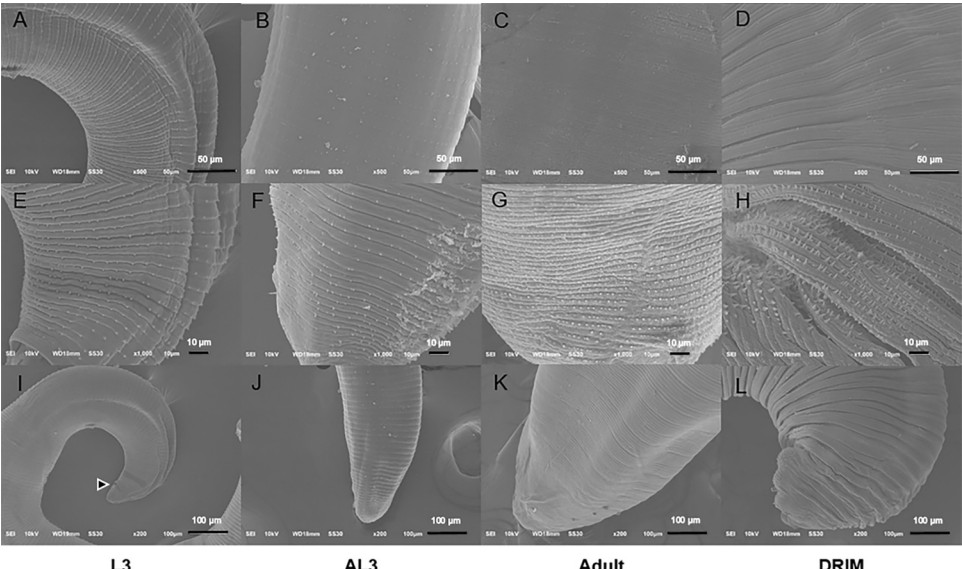

**Fig 4. Electron micrographs of the caudal part of *G. spinigerum* in any stage.** Caudal area (A-D); Cuticular spine with higher magnification (E-H); tail (I-L); L3 (A, E, I); AL3 (B, F, J); adult (C, G, K); STIM (D, H, L).

were seen for body cuticular spines along the transverse striations at the caudal end (**Table 2**). Body cuticular spines all presented single-pointed ends but differed in size between L3 and AL3 specimens, which were dense on the anterior third of the body and became gradually smaller in number and size with wide dispersion towards the posterior extremity. The STIM thus remained in a pre-adult stage where pseudobursae were not yet developed or expanded at the posterior extremity. A wide terminal anal pore (arrow head) was crescent-shaped and striated, with a denuded cuticle with tiny unidentate spines, measuring 36.52 to 43.01 μm in diameter on the ventral surface of L3 with transverse striations above the pore. These structures were also expected to be present ventrally in other stages since parasites develop inside the first intermediate host (cyclops) [8] (**Fig 4I**). The tail was bluntly round in all stages.

## Discussion

*G. spinigerum* larval worms from patients with superficial skin infestations that survive during albendazole administration are extremely rare. Such treatment typically kills worms, as previously reported in infected mice with AL3 [23,24]. Further, albendazole is an effective treatment for larvae lodged in human organs [25]. For this study, five years, from 2015 to 2020, passed before rarely three surviving *G. spinigerum* immature young adults were incidentally collected from patients. These patients expressed concern about their clinical symptoms after an appropriate time had elapsed for surviving worms to migrate to the skin. Patient treatments with albendazole are summarised in the Table 1. In case, treatment was successful, and gnathostome worms became surrounded by eosinophilic infiltration, then degenerated. Histologically, eosinophilic granulomas containing gnathostome larvae showed surrounding inflammation in necropsy tissue due to antigenic activity contributing to worm destruction [26].

From the STIM detected, the rows of cephalic spines increased from the original four transverse rows in L3 to eight rows in our study; Eight to 10 rows are typical for adults [8,27,28]. No extra row of cephalic spines was seen in AL3 from the infected rat due to the host's short development duration, as previously described [29]. The curved line on the head bulb of the L3 specimen showed a smoother, obtuse angle accompanied by continuous thick, well-developed oblong-shaped single-pointed spines in rows of more than 40 cephalic spines (**Fig 3A**). The length of each cephalic spine was greater than in AL3, STIM and adult stages. The latter group had more cephalic spines that developed later and appeared sharp-pointed and backwardly directed in rows with a similar claw-like shape (**Fig 3B–3D**). In a row-by-row comparison, the cephalic spine average length for L3 in the 1st to 4th row was greater than for AL3, STIM and the fully grown adult ($p$-value $< 0.05$). The cephalic spine length for the fully-grown adult in the 5th to 8th rows was greater than for the STIM ($p$-value $< 0.05$) as the STIM was still in juvenile form. The explanation might involve adaptation for developing from L3 into adult stages and the latter displays an increased number of cephalic spines and rows.

The most striking changes in the integument surface between the STIM and other stages were the length of unidentated, cuticular spines at the caudal extremity, including the lining pattern of body cuticular spines along the anterior one-third beginning next to the head bulb which was crooked and wicked, while the body cuticular spines of *G. spinigerum* adults are larger and more smoothly curved [30] (**Fig 5**). The integument surface of the fully-grown adult worm showed a clearly defined pattern of straight cuticular spines in an orderly arrangement. This finding was similar to previous reports of a *G. spinigerum* pre-adult worm incidentally collected from a patient during a segmental resection in the cecal wall and an immature adult collected from a surgical wound in the terminal ileum of another patient. The integument surface of these gnathostome worms were not exposed to any medical treatments and were different from the STIM [31,32].

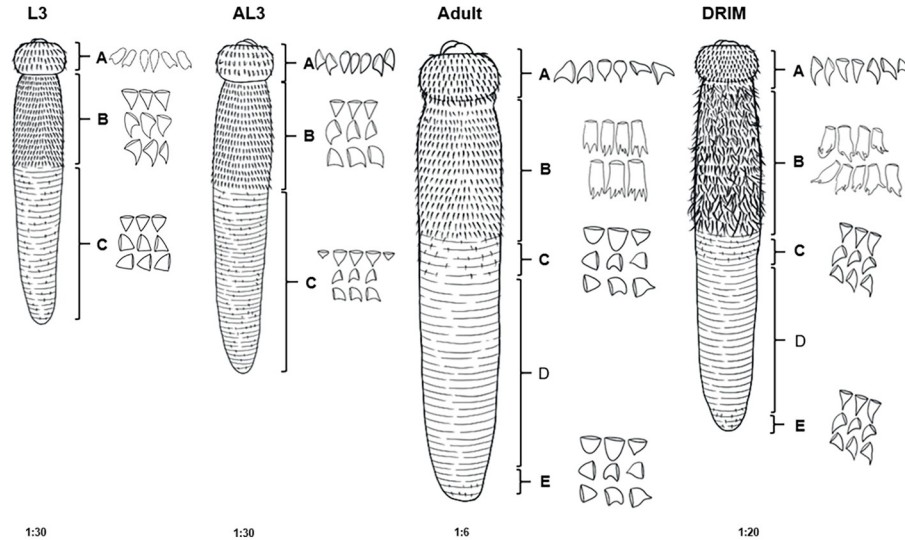

**Fig 5. Diagram for various stages of *G. spinigerum*.** Cephalic spines (A); Anterior body cuticular spines (B); Middle-Posterior body cuticular spines (C); Cuticular spine naked-areas (D); Caudal body cuticular spines (E). The ratio of real image: The ratio of magnification image (+).

This study demonstrated that the surface texture of the STIM worm integument remained in good condition with no structural damage, even though the STIM was exposed to albendazole. This drug induces cuticular alterations on adults of *Toxocara canis* adults evaluated using SEM [6]. Albendazole can cause malfunction of intestinal cells of gnathostomes by binding the colchicine-sensitive site on tubulin and inhibiting polymerisation of microtubules. The resulting lack of cytoplasmic microtubules impairs glucose uptake and glycogen deposition in both larval and adult stages. Albendazaole induced significant immobilisation after *in vitro* exposure of AL3 forms and caused some changes in larval cuticles [33,34]. Also, substantial tissue damage due to degenerative changes in muscular cells and fibres of contractile regions along with reduction of mitrochondria in the *G. binucleatum* AL3 stage were seen *in vitro* after exposure to albendazole [35]. However, effects seen *in vitro* are not necessarily reflective of what may occur clinically. Hence, albendazole doses of either 60 or 90 mg/kg were administered to experimentally infected mice and produced damage to the body walls of the *G. spinigerum* worms *in vivo*, especially in non-contractile areas of the muscular layer. A decrease in the number of normal mitochondria and an increase in abnormal mitochondria also occurred. Abnormal mitochondria developed large vacuoles and became distorted and degenerated [23]. Moreover, a previous report recommended albendazole (400 mg) before making incisions into skin lesions to increase the chance of eliminating the gnathostome-larval worms [36]. Gnathostome larvae were sometimes expelled after treatment with albendazole, but were internally destroyed or dead [18,25].

In this study, the changes of surface integument might possibly be involved with the survival albendazole-treated immature stage of *G. spinigerum* from the observed details appeared in the larvae detected. However, there is no exact survival rate reported in the human body once patients come back to see doctors with the same symptoms of migratory swelling and itching inflammation. The challenge of re-treatments with albendazole for another course or switch to ivermectin for alternative medicine has been practiced in the Parasite Clinic, Hospital for Tropical Diseases, Faculty of Tropical Medicine, Mahidol University. It is rather due to the failure of treatment more than the re-infection of the disease as patients do usually come

back within 3 months with the same symptoms [9,16]. Further study about albendazole treatment affecting the survival drug-treated larval stage is still challenging as the molecular information of more specimens is required to be specimens to be strongly significant confident to confirm this evidence in further work. However, the immature adult stage is not developed in the general used-experimental animal such as Wistar rat and mouse. Using the kind of other mammal hosts, cats and dogs, are the definitive host of this parasite not ethically allowed in Thailand which is the limitation of the study.

## Conclusions

Our findings are the first detailed observations of integument surfaces of a surviving *G. spinigerum* larvae following albendazole treatment. The findings contrast with previous studies of albendazole effects where no worm deformities were reported. Changes on the STIM specimen's integument surface included increased length of cuticular spines caudally even though the helminth was still in a pre-adult stage. The STIM also displayed a crooked and wicked lining pattern of cuticular spines with velvety cuticular folds anteriorly. These changes may assist the understanding of treatment resistance and survival of *G. spinigerum* immature young adults in the human body.

## Supporting information

**S1 File.**
(XLSX)

## Acknowledgments

This study was additional supported for journal publication by the Faculty of Tropical Medicine, Mahidol University. The authors would like to thank Dr. Paron Dekumyoy, Dr. Wallop Pakdee, Dr. Tippayarat Yoonuan for labour-intensive data support. Appreciated thanks are extended to the ENAGO for editing English grammar in the manuscript.

## Author Contributions

**Conceptualization:** Dorn Watthanakulpanich.

**Data curation:** Sumate Ampawong.

**Formal analysis:** Tapanee Kanjanapruthipong, Sumate Ampawong, Urusa Thaenkham, Dorn Watthanakulpanich.

**Investigation:** Tapanee Kanjanapruthipong, Sumate Ampawong, Urusa Thaenkham, Khwanchanok Tuentam, Dorn Watthanakulpanich.

**Project administration:** Dorn Watthanakulpanich.

**Resources:** Tapanee Kanjanapruthipong, Sumate Ampawong, Urusa Thaenkham, Khwanchanok Tuentam.

**Supervision:** Sumate Ampawong.

**Writing – original draft:** Tapanee Kanjanapruthipong, Sumate Ampawong, Urusa Thaenkham, Khwanchanok Tuentam, Dorn Watthanakulpanich.

**Writing – review & editing:** Tapanee Kanjanapruthipong, Sumate Ampawong, Urusa Thaenkham, Khwanchanok Tuentam, Dorn Watthanakulpanich.

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
