## [Decision Letter · Decision Letter 0]

29 Nov 2021

PONE-D-21-35504Survival of immature pre-adult Gnathostoma spinigerum in humans after treatment with albendazolePLOS ONE

Dear Dr. Watthanakulpanich,

Thank you for submitting your manuscript to PLOS ONE. After careful consideration, we feel that it has merit but does not fully meet PLOS ONE’s publication criteria as it currently stands. Therefore, we invite you to submit a revised version of the manuscript that addresses the points raised during the review process.

ACADEMIC EDITOR:

#Reviewer 1

In this manuscript, the authors first reported detailed observations of integument surfaces of a surviving G. spinigerum larvae in patients following albendazole treatment, which no worm deformities of albendazole effects were reported previously. The manuscript has merit because the provided data is important and the results are helpfully referred to using albendazole treatment of G. spinigerum for clinical patients. Nevertheless, the manuscript still contains some problems need to be addressed before it is accepted. These comments are revealed as below.

Comments follow:

1.     First of all, the author conclude the DRIM of G.spinigerum have changed phenotypes of their integument surfaces, but it is a pity about the author miss to discuss how these changes that may affect the effects of G. spinigerum immature adults in the albendazole treatment resistance and survival rate in the human body; or how these treatments affect the phenotype changes of drug-resistant G. spinigerum immature adults. Although the author mentioned the possible effects of albendazole on AL3 or worm adults body morphology in the discussion, whether there is a similar situation in DRIM is the focus. These issues should be included in the discussion.

2.     Introduction, how many cases be reported in Thailand currently?

3.     Line 71, “Relapses are attributed to treatment failure rather than reinfection.” This description should add a source of reference.

4.     Material and Methods, Line 118, this part of the description should be more detailed. Does the worm adults refer to be taken from a dog? What kind of dog is it? How old is it?

5.     Results, the parts specifically described or pointed out on the figure can be pointed out with arrows, so that the reader can more clearly know where the author wants to express the key points. In the first paragraph of the results, the part about Figure 3 should be moved to the second paragraph about the SEM study.

6.     Table 2. Significant differences should be marked directly in the table¸ such as * and the p value.

7.     Line 174-175, there is an error in the description of "The head spines were counted and measured under SEM to compare with the control (Table 1)”. Here Table 1 shows "Clinical data for three female gnathostomiasis patients who were the source of three recovered DRIM", the data is missing, should revise.

8.     Many parts of the second and third paragraphs in the discussion of text should be described in the results, and there is no need to describe the observations again in the discussion.

9.     All relevant graphics and drawings should be dimensioned, and Figure 1 can be condensed into one page.(less...)

#Reviewer 2

This paper is very interesting to find the possible clues for the survival of immature pre-adult Gnathostoma spinigerum in humans after treatment with albendazole, addressing the difference in morphology among larval, juveniles, adult and drug-resistant one employing SEM. Although they have important findings, a question is that how can the authors determine due to difference in morphology is the reason for this worm to be drug-resistant? With no molecular assessment, it's very hard to say this kind of worm is drug-resistant. This point should be addressed by the authors.

A marked-up copy of your manuscript that highlights changes made to the original version. You should upload this as a separate file labeled 'Revised Manuscript with Track Changes'.An unmarked version of your revised paper without tracked changes. You should upload this as a separate file labeled 'Manuscript'.

We look forward to receiving your revised manuscript.

Kind regards,

Chia Kwung Fan, LL.M, PhD

Academic Editor

PLOS ONE

Journal Requirements:

2. As part of your revision, please complete and submit a copy of the Full ARRIVE 2.0 Guidelines checklist, a document that aims to improve experimental reporting and reproducibility of animal studies for purposes of post-publication data analysis and reproducibility: https://arriveguidelines.org/sites/arrive/files/Author%20Checklist%20-%20Full.pdf (PDF). Please include your completed checklist as a Supporting Information file. Note that if your paper is accepted for publication, this checklist will be published as part of your article.

(This study was financially supported by the Faculty of Tropical Medicine, Mahidol University.)

(The author(s) received no specific funding for this work.)

Reviewers' comments:

Reviewer's Responses to Questions

**Comments to the Author**

1. Is the manuscript technically sound, and do the data support the conclusions?

Reviewer #1: Yes

Reviewer #2: Yes

2. Has the statistical analysis been performed appropriately and rigorously? 

Reviewer #1: I Don't Know

Reviewer #2: Yes

3. Have the authors made all data underlying the findings in their manuscript fully available?

Reviewer #1: Yes

Reviewer #2: Yes

4. Is the manuscript presented in an intelligible fashion and written in standard English?

Reviewer #1: Yes

Reviewer #2: No

5. Review Comments to the Author

Reviewer #1: In this manuscript, the authors first reported detailed observations of integument surfaces of a surviving G. spinigerum larvae in patients following albendazole treatment, which no worm deformities of albendazole effects were reported previously. The manuscript has merit because the provided data is important and the results are helpfully referred to using albendazole treatment of G. spinigerum for clinical patients. Nevertheless, the manuscript still contains some problems need to be addressed before it is accepted. These comments are revealed as below.

Comments follow:

1. First of all, the author conclude the DRIM of G.spinigerum have changed phenotypes of their integument surfaces, but it is a pity about the author miss to discuss how these changes that may affect the effects of G. spinigerum immature adults in the albendazole treatment resistance and survival rate in the human body; or how these treatments affect the phenotype changes of drug-resistant G. spinigerum immature adults. Although the author mentioned the possible effects of albendazole on AL3 or worm adults body morphology in the discussion, whether there is a similar situation in DRIM is the focus. These issues should be included in the discussion.

2. Introduction, how many cases be reported in Thailand currently?

3. Line 71, “Relapses are attributed to treatment failure rather than reinfection.” This description should add a source of reference.

4. Material and Methods, Line 118, this part of the description should be more detailed. Does the worm adults refer to be taken from a dog? What kind of dog is it? How old is it?

5. Results, the parts specifically described or pointed out on the figure can be pointed out with arrows, so that the reader can more clearly know where the author wants to express the key points. In the first paragraph of the results, the part about Figure 3 should be moved to the second paragraph about the SEM study.

6. Table 2. Significant differences should be marked directly in the table¸ such as * and the p value.

7. Line 174-175, there is an error in the description of "The head spines were counted and measured under SEM to compare with the control (Table 1)”. Here Table 1 shows "Clinical data for three female gnathostomiasis patients who were the source of three recovered DRIM", the data is missing, should revise.

8. Many parts of the second and third paragraphs in the discussion of text should be described in the results, and there is no need to describe the observations again in the discussion.

9. All relevant graphics and drawings should be dimensioned, and Figure 1 can be condensed into one page.

Reviewer #2: This paper is very interesting to find the possible clues for the survival of immature pre-adult Gnathostoma spinigerum in humans after treatment with albendazole, addressing the difference in morphology among larval, juveniles, adult and drug-resistant one employing SEM. Although they have important findings, a question is that how can the authors determine due to difference in morphology is the reason for this worm to be drug-resistant? With no molecular assessment, it's very hard to say this kind of worm is drug-resistant. This point should be addressed by the authors.

6. PLOS authors have the option to publish the peer review history of their article (what does this mean?). If published, this will include your full peer review and any attached files.

Reviewer #1: No

Reviewer #2: No

---

## [Author Response · Author response to Decision Letter 0]

13 Jan 2022

Editor and reviewer's comments are useful for improving the manuscript

---

## [Decision Letter · Decision Letter 1]

17 Feb 2022

Survival of immature pre-adult Gnathostoma spinigerum in humans after treatment with albendazole

PONE-D-21-35504R1

Dear Dr. Watthanakulpanich,

We’re pleased to inform you that your manuscript has been judged scientifically suitable for publication and will be formally accepted for publication once it meets all outstanding technical requirements.

Kind regards,

Chia Kwung Fan, LL.M, PhD

Academic Editor

PLOS ONE

Additional Editor Comments (optional):

Reviewers' comments:

Reviewer's Responses to Questions

**Comments to the Author**

1. If the authors have adequately addressed your comments raised in a previous round of review and you feel that this manuscript is now acceptable for publication, you may indicate that here to bypass the “Comments to the Author” section, enter your conflict of interest statement in the “Confidential to Editor” section, and submit your "Accept" recommendation.

Reviewer #1: All comments have been addressed

Reviewer #2: All comments have been addressed

2. Is the manuscript technically sound, and do the data support the conclusions?

Reviewer #1: Yes

Reviewer #2: Yes

3. Has the statistical analysis been performed appropriately and rigorously? 

Reviewer #1: Yes

Reviewer #2: Yes

4. Have the authors made all data underlying the findings in their manuscript fully available?

Reviewer #1: Yes

Reviewer #2: Yes

5. Is the manuscript presented in an intelligible fashion and written in standard English?

Reviewer #1: Yes

Reviewer #2: Yes

6. Review Comments to the Author

Reviewer #1: All the comments have been addressed well. I have no problem for this manuscript that is ready for publishing.

Reviewer #2: The authors have addressed appropriately the points raised by the reviewer thus the reviewer recommend it to be accepted for publication.

7. PLOS authors have the option to publish the peer review history of their article (what does this mean?). If published, this will include your full peer review and any attached files.

Reviewer #1: No

Reviewer #2: No

---

## [Editor Report · Acceptance letter]

28 Feb 2022

PONE-D-21-35504R1 

Survival of immature pre-adult *Gnathostoma spinigerum* in humans after treatment with albendazole 

Dear Dr. Watthanakulpanich:

I'm pleased to inform you that your manuscript has been deemed suitable for publication in PLOS ONE. Congratulations! Your manuscript is now with our production department. 

Kind regards, 

on behalf of

Dr. Chia Kwung Fan 

Academic Editor

PLOS ONE